# What challenges did junior doctors face while working during the COVID-19 pandemic? A qualitative study

Johanna Spiers [ID],[1] Marta Buszewicz,[2] Carolyn Chew-Graham,[3] Alice Dunning,[4] Anna Kathryn Taylor [ID],[5] Anya Gopfert,[6] Maria Van Hove,[7] Kevin Rui-Han Teoh [ID],[8] Louis Appleby,[9] James Martin,[10] Ruth Riley [ID] [10]

**Correspondence to**
Dr Johanna Spiers;
johanna.spiers@gmail.com

## ABSTRACT

**Objectives** This paper reports findings exploring junior doctors' experiences of working during the COVID-19 pandemic in the UK.

**Design** Qualitative study using in-depth interviews with 15 junior doctors. Interviews were audio-recorded, transcribed, anonymised and imported into NVivo V.12 to facilitate data management. Data were analysed using reflexive thematic analysis.

**Setting** National Health Service (NHS) England.

**Participants** A purposive sample of 12 female and 3 male junior doctors who indicated severe depression and/or anxiety on the DASS-21 questionnaire or high suicidality on Paykel's measure were recruited. These doctors self-identified as having lived experience of distress due to their working conditions.

**Results** We report three major themes. First, the challenges of working during the COVID-19 pandemic, which were both personal and organisational. Personal challenges were characterised by helplessness and included the trauma of seeing many patients dying, fears about safety and being powerless to switch off. Work-related challenges revolved around change and uncertainty and included increasing workloads, decreasing staff numbers and negative impacts on relationships with colleagues and patients. The second theme was strategies for coping with the impact of COVID-19 on work, which were also both personal and organisational. Personal coping strategies, which appeared limited in their usefulness, were problem and emotion focused. Several participants appeared to have moved from coping towards learnt helplessness. Some organisations reacted to COVID-19 collaboratively and flexibly. Third, participants reported a positive impact of the COVID-19 pandemic on working practices, which included simplified new ways of working—such as consistent teams and longer rotations—as well as increased camaraderie and support.

**Conclusions** The trauma that junior doctors experienced while working during COVID-19 led to powerlessness and a reduction in the benefit of individual coping strategies. This may have resulted in feelings of resignation. We recommend that, postpandemic, junior doctors are assigned to consistent teams and offered ongoing support.

## Strengths and limitations of this study

► Participants were interviewed at the peak of the second wave of COVID-19 in the UK, meaning transcripts contain data that are highly relevant to the research question
► In-depth, reflexive thematic analysis was carried out on the data, leading to the development of rich, insightful themes.
► Female participants outnumbered male participants in this study, potentially leading to gender imbalance.
► Additionally, the wider study was not initially designed to explore experiences of working during COVID-19. Instead, participants naturally discussed this topic during interviews.

## INTRODUCTION

Doctors are more vulnerable to mental illnesses (such as anxiety and depression) and suicide than the general population.[1 2] In recent years, including those before the COVID-19 pandemic, UK doctors have reported understaffing, stretched resources, increased workload and burnout.[3–7]

There is an additional need to attend to frontline workers' well-being during health crises.[6–8] Frontline workers caring for patients with COVID-19 have reported stress and distress due to the strain on healthcare systems.[9] Such stressors include the need for rapid training around treating a new illness[9] and the psychological impact of exposure to unprecedented levels of suffering and COVID-19 related deaths, both of patients and colleagues.[8 10 11]

These stressors led to healthcare professionals (HCPs) reporting fears about contracting or spreading the virus as well as uncertainty due to new ways of working.[11 12] Impacts of these fears and stressors include reduced sleep, self-harm, panic attacks, guilt, relationship breakdowns,[11] concerns about lack of training[7] and psychological trauma.[10]

The UK reported some of the highest numbers of COVID-19 cases in Europe.[7] In a recent paper, almost half of the 224 UK doctors surveyed (from junior doctors to consultants) felt that their mental health had been harmed by the pandemic, while a third reported impacts to their physical health.[5] Increased healthcare worker burnout is, therefore, a major concern at this time. We need a holistic understanding of the experiences and needs of frontline workers to mitigate psychological distress and burnout.[11]

'Junior doctor' is the term given to qualified doctors who are still in training while working. They may have 8 or more years of experience, depending on their specialty.[13] Junior doctors have reported fears that they will 'fail' or appear 'weak' if they take time off sick, making it harder for them to report mental health concerns.[14] This group faced unique challenges during COVID-19 due to uncertainties about exams,[6] potential redeployment[8 15 16] and concerns about their learning opportunities.[15 16] UK junior doctors have reported that they did not receive enough education before treating patients with COVID-19.[15] They were also often faced with the difficult task of contacting patients' families to provide updates, since relatives were typically not permitted to visit.[15]

Despite this, few researchers have looked in-depth at the psychological experiences of junior doctors. Instead, they have explored practical matters relating to this group, such as the resilience of new rotas (ie, assigning enough staff to cope with the workload),[17] redeployment,[15 16] the impact on training[18] and the provision of certain services such as obstetrics and gynaecology.[19]

Researchers have posited the need for more in-depth qualitative analysis in this area.[5 11] This paper is part of a wider study[20 21] designed to explore the impact of working conditions and cultures on junior doctors in general. Data collection coincided with the second wave of the pandemic in the UK, meaning the topic naturally arose for all 15 participants interviewed. As such, we aim to address this crucial gap in the literature and reflect the experiences of junior doctors working within the context of COVID-19.

## METHOD
### Study design and setting
This qualitative study is part of a larger mixed-methods study exploring junior doctors' perceptions of stress and distress. Semistructured interviews were used to explore junior doctors' experiences of working during COVID-19. The study setting was the National Health Service (NHS) in England.

### Sampling and recruitment
A total of 456 junior doctors were initially recruited for an online survey exploring working cultures, psychological distress and suicidality between November 2020 and March 2021. They self-identified as participants, accessing the survey through posts on social media, junior doctor forums and via emails sent from their specialty schools. Survey participants whose results indicated severe depression and/or anxiety on the Depression Anxiety and Stress Scales (DASS-21) questionnaire[22] or high suicidality on Paykel's measure[23] were contacted via email to ask if they would like to take part in an in-depth, qualitative survey. As such, it should be noted that, in line with our qualitative methodology, this was a small, purposive sample of junior doctors who were experiencing stress and distress as a result of their working conditions. Thus, findings cannot be generalised to all junior doctors. However, it should also be noted that levels of distress were high in the whole sample of surveyed junior doctors. A total of 27 potential participants were contacted, of which 15 were female, 9 male and 3 undisclosed.

Interested individuals contacted JS and gave informed consent. Participants were given the chance to ask JS questions about the research team and the study before interviews went ahead. Fifteen junior doctors (12 female, 3 male) were recruited.

### Data collection
A semistructured interview guide was developed by the research team and modified iteratively as data collection and analysis progressed. This guide aimed to capture participants' views, experiences, feelings and beliefs about working conditions and cultures that were perceived as stressful or distressing. The guide was informed by the existing literature,[1 3 14] input from junior doctors on the study team as well as patient and public involvement (PPI) consultation exercises conducted before obtaining funding. Following conventions for semistructured interviews,[24] points from the topic guide were followed up with individualised questions exploring topics of interest and importance for each participant.

Interviews were conducted either on the telephone or via video call, from participants' homes or places of work. They took place between December 2020 and February 2021—that is, during the second wave of the COVID-19 pandemic in the UK—and at a date and time that were convenient to the participants. A risk protocol was used to ensure that appropriate support from two senior General Practitioners (GPs) who were on the study team and/or practitioner health would be provided to participants in the event of the disclosure of suicidal ideation. The in-depth interviews were conducted by JS, a female PhD psychologist with extensive qualitative methods expertise. JS also recorded any pertinent observations in field notes following each interview. Interviews ranged from 29 min to 102 min in length (mean=62.8 min).

The audio-recorded interviews were transcribed verbatim and checked for accuracy by JS before analysis. All transcripts were anonymised before discussion within the wider research team. Reflexive notes were recorded by researchers throughout the process.

## Patient and public involvement and engagement

There are three junior doctors on the research team, all of whom consulted with other colleagues in the PPI team about the initial research idea and participated in analysis meetings. Five junior doctors gave feedback on the initial funding application, while four fed back on the protocol, topic guide and participant-facing documents.

## Data analysis

Data were analysed by JS using reflexive thematic analysis,[25 26] in which themes highlight patterns of shared meaning united by a core concept. An inductive, explicit, critical realist approach was adopted since this was in line with the researchers' desire for a rich, data-driven analysis that demonstrated the interplay between events and participants' interpretations of those events.[25] Data saturation is not a relevant concept within this type of approach, in which it is accepted that each new participant adds fresh insights. Analysis began once all interviews had been conducted. Transcripts were analysed one by one using NVivo V.12. As analysis progressed, a table of themes was generated and refined. Each new transcript led to new codes and themes being added or expanded. In addition, four members of the team, one of whom was a junior doctor and two of whom were academic GPs (RR, MB, AT and CC-G), read and fed back on 6 of the 15 interviews. Their views and insights were collaboratively incorporated into the NVivo codes. JS then refined these codes to create relevant tables of themes once all interviews had been analysed and discussed. Analysis continued and deepened during the write-up, where shared meanings were generated and described for each theme.[26]

## Reflexivity

RR, the Principle Investigator (PI), is epistemologically steeped in qualitative traditions underpinned by interpretivism and phenomenology, and is oriented by critical theory such as feminism. This is likely to have influenced her interest in exploring why female doctors are more likely to experience distress.

JS, the lead analyst on this paper, is a qualitative health psychologist. She has an interest in in-depth, interpretative methods. She is white, cis-gendered, heterosexual and able-bodied. This heteronormative positioning is likely to have impacted her interviewing and analysis.

Both researchers have an interest in the systemic issues impacting individual NHS workers and are motivated by trying to find organisational—rather than individual—solutions for those workers.

The junior doctor (AKT) and academic GPs (MB and CC-G) who also contributed to analysis of the data have experienced and observed events during their professional lives that may have influenced how strongly they interpreted the data. Additionally, MB and CC-G have a strong interest in mental health.

## Findings

All 15 participants discussed the impact of COVID-19 on their working conditions. Findings divided into three major themes: challenges of working during the COVID-19 pandemic; strategies for coping with the impact of COVID-19 on work; and positive impact of the COVID-19 pandemic on working practices.

See table 1 for an overview of all relevant themes and subthemes.

### Challenges of working during the COVID-19 pandemic

Participants described challenges related to their work as junior doctors during the COVID-19 pandemic. Challenges were personal or work related.

#### Personal impact: helplessness in the face of trauma

Working as junior doctors during the COVID-19 pandemic affected participants' mental health. Throughout this theme, there is a sense that participants felt helpless and powerless as they strove to carry out their jobs in such unmapped territories.

One participant described the harmful impact of being exposed to death and suffering:

**Table 1** Table of themes for junior doctors' experience of working during the COVID-19 pandemic

| Theme | Subtheme |
|---|---|
| Challenges of working during the COVID-19 pandemic *'[P]atients were just dying in front of us so quickly and they were young'*. (P5) | Personal impact: helplessness in the face of trauma *'[M]y sleep is awful again, I'm waking up, I think Covid-19 hasn't helped'*. (P1) |
| | Work-related impact: change and uncertainty *'I gained 14 new patients who I'd not met before'*. (P6) |
| Strategies for coping with the impact of COVID-19 on work *'So although I should have moved on from GP I ended up staying in GP so I was actually there for eight months'*. (P7) | Limitations of personal strategies *'I cried a lot'*. (P14) |
| | Organisational strategies *'[W]e'd kind of share what we've learned'*. (P5) |
| Positive impact of COVID-19 on working practices *'[S]ince Covid-19, things have improved slightly there's, um we have something called like the rest and recuperation hub'*. (P6) | Positive new ways of working *'We almost looked forward to going to work'*. (P8) |
| | Additional support and camaraderie *'[T]hey provided hot meals'*. (P5) |

I'd seen (pause) a whole ward just emptied out and then refilled overnight, after people had just died. It was horrendous. Uh, I was like, 'I need to talk to somebody about this or I'm just going to go home and cry'. (P5, female)

This participant's language—'emptied out' and 'refilled'—suggests that the patients with COVID-19 had become dehumanised for her; a mass of unwell bodies who were dying and being replaced in an almost mechanical manner. She was helpless to stop this flow of nameless bodies.

Participants felt helpless in the face of fears for their own safety and that of their loved ones. Initially, they were unsure of how to protect themselves or of the risk they might pose to their families:

… we had someone that we thought was, um, COVID-19, but it was very, very early on. And I remember being told off for wearing a mask. (P3, female)

… we were worried about if we were taking home our clothes, if we were making other people sick, if we would get sick, it was an incredibly stressful working environment. (P5, female)

As time went on, fears for personal safety came from different sources, with one participant reporting that her colleagues were not maintaining safety standards. However, as a junior doctor, she felt powerless to ask for this to change:

It's not patients, it's staff. I find that really stressful. Like you walk past an office and there might be two or three people sat in an office having a chat, all with their masks under their chin. […] I don't feel confident enough to knock on the window and be like guys, what are you doing? But I know that them doing it puts me more at risk and puts the patients at risk. […] You see stuff being wrong and you're like every day, like multiple times a day you're like do I say something, do I not say something? And you feel bad for not saying something. (P10, female)

This description of discomfort could be defined as moral injury; that is, the distress that occurs when a person witnesses or carries out an act that is contrary to their values. The participant felt uncomfortable and helpless however she responded.

Participants were powerless to switch off or rest when they got home from work:

You couldn't switch off, you never felt like you'd had, uh, done a good job. (P5, female)

… my sleep is awful again, I'm waking up, I think COVID-19 hasn't helped with these sort of flashbacks. (P1, female)

Participants did not feel clinically supported in the new working environment caused by COVID-19, which led to further helplessness, fear and trauma. The lack of support could be practical:

I'm going to personally take responsibility for changing […] the big scary machine that I'm not trained in, and, uh, figure out how it works, while the patient is there trying to physically die in front of me, but so are five others, so oh well, no help is coming. (P5, female)

Some felt unsupported psychologically, with one participant appearing to feel that her needs were invisible to those who might support her:

… they got some psychologists who would be available and very occasionally they would come on the ward. (pause) And they would talk to the nurses. And that was it. No. It felt assumed to be on the nurses and people working in ITU and just ordinary junior doctors didn't (pause) didn't seem to matter. (P14, female)

Another felt unsupported in terms of her physical health; her safety was compromised, meaning she was unable to protect herself:

… you will turn up on a ward and you will find out halfway through handover that they've had a positive case over the weekend. (P10, female)

Additionally, a junior doctor whose family were overseas reported feeling unsupported by her hospital after contracting COVID-19:

… when I went to quarantine, I realised that no-one actually cares about you from the hospital? […] No-one called me! […] When I was very very sick, imagine that, if I had, if I had literally no-one. (P4, female)

### Work-related impact: change and uncertainty

The work-related impact of working during a COVID-19 context centred around uncertainty and change. These included changes to workload, staffing levels, relationships with colleagues and patients, lack of support and uncertainty around new ways of working.

The junior doctors' workload grew significantly when COVID-19 hit, leading to further stressors. This led to a huge and stressful increase in one participant's responsibilities:

… on a Friday in the middle of the day when there was no consultant around […] I gained 14 new patients who I'd not met before […] that was a really stressful day. (P6, female)

Workload increased out of hours as well, as participants were constantly having to learn new facts about the virus and its management. The quote below demonstrates the doubt and pressure felt while trying to learn in the face of unmanageable amounts of new information:

So we were getting 20 emails a day, and every single one would have a red flag saying 'vital, important, must read', and you'd worry you'd missed something […] there's so much information, it was constant,

and you couldn't switch off, because it would impact your job. (P5, female)

As workload rose, staffing levels, which had already been stretched, were adversely affected by further staff reductions due to illness or the need to self-isolate, demonstrating additional change and uncertainty:

So it's very very short-staffed because a lot of the people are self-isolating, ill with COVID, or just because you know they've worked already five or six days in a row, and obviously they're quite tired and they have to take a break. (P12, male)

The additional workload changed working relationships in various ways. Participants reported that colleagues became irritable or verbally aggressive due to increased stress:

I think everyone got a little bit more [pause], um, maybe snippy? With each other? 'Cause we were all are very stressed and anxious. (P3, female)

… a registrar wearing an MF53 mask (that is, a full face, military style of mask) and the consultant laying into him basically shouting at him that […] he was depriving someone else who actually needed this mask […] emotions were running high because people were scared. (P14, female)

One trainee, based in general practice, reported that patients had become abusive during telephone appointments, potentially dehumanising their doctors:

… sometimes people lose sense of the fact that it's another human being on the end of the phone with them. And you're already dehumanised a little bit as a doctor because people expect you to be more than, more than human. And when you then couple that with someone just being this kind of like faceless voice on the end of the phone, especially when people are scared or something like that, it just there's that heightened level of aggression. (P7, female)

That participant also reported finding the change to telephone appointments clinically challenging and risky in terms of being able to diagnose patients accurately.

I don't think you realise how much you rely on seeing someone in front of you to know how well they are. And talking to someone over the phone it just feels a lot more dangerous. (P7, female)

Compounding these changes that made participants' working lives harder was the fact that it also became harder to speak with and get support from peers due to the safety measures:

Um, and now with COVID where you're only allowed, like, two people in each room, it, it's very difficult to, um, socialise and talk. (P8, female)

As junior doctors in training, participants also found the uncertainty around changes to rotas and exams challenging:

… a fair amount of uncertainty and the problem this time is that, ah, a lot of courses are still going ahead, exams are still going ahead, but we've been moved onto emergency rotas with a week's notice. (P8, female)

The junior doctors were often expected to work in different specialities or locations from those which they had been allocated to prepandemic. Constant anxiety due to uncertainty about redeployment was reported:

… anxious and uncertain about whether that was going to happen and would sort of check my emails pretty consistently to see whether that was actually whether that was going to be um delayed or stopped because of COVID redeployment. (P6, female)

The pandemic meant that new ways of working were quickly developed and implemented. Trying to adjust to these changes was another challenge. One trainee in psychiatry talked about the potential stress and impact on patient care of working from home:

… you've not got those people around you to bounce things off, so you might get an email and it might be quite an anxiety-inducing email because it might say someone's suicidal, you need to see them, and you're thinking, oh, I can't see them, erm, and normally, kind of in an office you'd just be able to ask, can anyone else see them? (P2, female)

### Strategies for coping with the impact of COVID-19 on work

Participants described both personal and organisational strategies for coping with the above challenges.

#### Inadequate personal coping strategies

Emotion-focused and problem-focused coping strategies were used for dealing with the challenges of COVID-19. However, there was a sense that these personal coping strategies, which might have been adequate before COVID-19, were not enough to protect participants from the impact of working during the pandemic.

Emotion-based coping strategies included crying:

So I cried a lot outside. Because it was getting warmer so you could go outside. Hug a tree, cry. (P14, female)

Stoicism was used by some, although this latter strategy suggested a sense of resignation, illustrated by P8's rhetorical and hopeless question:

And [pause] and in a way it didn't really matter that our rota changed, because there was nothing else to do? (P8, female)

A sense of powerlessness combined with acceptance was perceived to have impacted the profession as a whole:

I think erm you know everyone's a bit more sort of resigned to things now and it feels like we've sort of erm entered a collective sort of depressive state of acceptance. (P9, male)

We can see that these personal, emotion-based coping strategies had their limits when employed during the COVID-19 pandemic.

Problem-focused strategies were perhaps more effective. One participant volunteered to take on the work of calling relatives to let them know their loved ones were very sick, perhaps to be able to provide a more personal input to such a traumatic situation.

I used to volunteer to kind of be the person making those phone calls, cos it was, it felt like you were able to do something about it at least? It wasn't that sort of like, 'I put lines in people and hopefully', and then just watching them die. (P5, female)

Another participant agentically took control of her situation by arranging more support for herself, perhaps in response to the helplessness described in the previous theme:

… when lockdown came back in […] I noticed that like I was feeling low so I referred myself to the Let's Talk Wellbeing, erm which is like the community, CBT, GP, self-referral system. And I found that really helpful erm so that kind of stopped me spiralling. (P10, female)

### *Organisational strategies*
Just as participants found ways to cope with the challenges of COVID-19, so did the organisations and teams for whom they worked, with some trusts and teams demonstrating collaborative, flexible thinking. One participant reported flexibility in terms of working from home for colleagues who had to self-isolate:

… most of the places have let the person sort of choose whether they you know, if it's your child that's got a fever and actually you know you're isolating and could do things then that's fine. But if you're poorly then you're poorly and that's fine. (P11, female)

Another described the need for her team to make pragmatic decisions about how to treat patients with COVID-19:

So if someone was clearly dying, they would [pause] be stepped down to a normal ward because on a normal ward they could at least have a visitor for one hour a day. (P14, female)

A third participant reported that her team pulled together to help one another in the new circumstances:

… there'd be so many [emails] even coming in during our shift, we'd divide it up, so we'd say, 'you read these five, I'll read these five, you read these five,

and then I'll read these five', and then we'd kind of share what we've learned from them. (P5, female)

### Positive impact of COVID-19 on working practices
Participants reported that working as junior doctors during the COVID-19 pandemic had some positive impacts. These included new and less bureaucratic ways of working as well as additional support and camaraderie.

### *Positive new ways of working*
Positive changes revolved around a less bureaucratic way of working, which included consistent teams, longer rotations and less red tape.

Several participants reported that they were now working in a consistent team, rather than regularly working with new colleagues. This was experienced as positive:

So normally, you're kind of working with somebody new every day almost. But we worked in teams that didn't rotate, so you had […] this team that you worked with very intensely for those four months as well, and that support structure was really good. (P5, female)

… we got really to know each other, we had a little social WhatsApp group where we'd, like, post pictures of the cakes we were gonna bring in, you know, everyone bought in food. We almost looked forward to going to work because you were like, oh, my buddies are there. (P8, female)

A sense of being part of a team and able to enjoy work comes across in the previous quote, where cake and conversation bring some positivity to the bleak picture painted thus far.

Rotations were paused for many junior doctors. Although this could lead to uncertainty, as reported in an earlier theme, it also had some potentially positive impacts:

So we were on the first rotation for four months and then the second for eight months […] Um, so, I guess it would have depended on what ward you got stuck on? [interviewer laughs] Um, I got stuck on one of the nice placements, I really enjoyed myself on the ward. (P3, female)

Various practical changes to working patterns were also experienced as positive. These included simple factors such as the ability to work from home and reduced red tape:

… just get away and do something relaxing, even if it's just go for a walk around the local canal and come back on a lunchtime is so much more achievable when you're working from home. So I think that's been really good. (P2, female)

… they say oh, we want you to travel to a hospital on your day off to show us your passport and your GMC certificate. And it's like I've been – doing this for 10

years. I've worked for you six times! Like, you've got my details (both laugh). And that's one thing where COVID has been really good because now they do it online and I'm like, why couldn't you have always done this? (P8, female)

One participant even appeared to cite COVID-19 as a motivator for returning to work at the NHS after time in another career:

… then COVID came and I wanted to come back to medicine anyway so I thought okay fine then let's just crack on with erm with the NHS. (P15, female)

### Additional support and camaraderie
Some participants reported that new supportive measures, such as additional facilities, had been put in place by their workplaces:

… they provided hot meals, which, at the beginning, when there were huge queues at the supermarket, and we were working 12-hour shifts, five days a week, and, um, [pause] and it was unpredictable whether you could kind of get food, because there were a lot of shortages and things. (P5, female)

And since COVID, things have improved slightly, there's, um we have something called like the rest and recuperation hub, which is like a room erm that does free teas and coffees and a few snacks […] you go there on your breaks to relax. (P6, female)

Another participant reported that her hospital made an effort to offer junior doctors support, although this was against the backdrop of a toxic working environment:

I'd say the culture's getting worse except for the fact that they send an e-mail out every now and again with some contact numbers [for support services] and that's what COVID has done. (P1, female)

It should be noted that the reports of improved support were tempered; note that participants reported 'slight' improvements to a culture that was also 'getting worse'.

### DISCUSSION
Fifteen distressed junior doctors were interviewed between December 2020 and February 2021 about their perceptions of stress and distress in their workplace cultures. All participants discussed how COVID-19 impacted their experiences. Looking at our themes as a gestalt, we suggest that the helplessness that arose due to the trauma of working during the pandemic meant that individual coping strategies that may have been more beneficial during less unusual times fell short, something that often went unrecognised by employers. To compound this, participants were also not sufficiently supported either practically or psychologically during this time. This may have led to feeling powerless and resigned in the face of difficult circumstances for which they were unprepared.

Additionally, we recommend that the positive lessons highlighted in this paper are followed in the long term.

Helplessness was commonly reported while working during COVID-19. Specifically, one participant described how traumatic it was to see so many patients dying. Others[8 10 27] have cited grief and managing such large numbers of patient deaths as especially challenging. We suggest that newer junior doctors might need extra support to process grief in such exceptional circumstances, for which they had not been trained. This might especially be the case for younger doctors[28] and female doctors,[1 29] since it has been shown that these groups, who made up the majority of participants in the current study, are potentially more vulnerable to depression, stress and suicidal thoughts.

Another participant reported an experience of moral injury following the unsafe behaviour of other staff members. Moral injury due to redeployment away from long-term patients[15] and concerns about letting patients down[8] during COVID-19 has also been reported. Our findings add an additional perspective, demonstrating that moral injury can also arise due to staff members neglecting safety protocols.

Adding to these traumatic personal experiences, participants reported that while their workload rose, staffing levels often decreased. Previous research has shown that, following austerity,[30] UK HCPs were already working in an under-resourced environment and that additional workload is a potent stressor.[21 31–33] Crises such as COVID-19 further emphasise the need for extra resources for our healthcare system, echoing the recommendations of the 2009 Boorman report,[34] which have been widely neglected.[6] It is often harder for frontline workers to take breaks during a pandemic,[5 10] adding to the potential for burnout since longer working hours are a risk factor.[29] Cubitt and colleagues[5] have highlighted the need for rotas that enable well-being rather than merely being resilient, that is, containing the bare minimum number of doctors per shift. Qualitative research such as the current study adds depth to these recommendations by demonstrating the instability, lack of support and powerlessness that distressed HCPs faced during this time.

Participants felt unsupported while working in these new, traumatic circumstances, a finding reflected elsewhere.[7 10 11] For example, one participant who needed psychological support intimated that she felt invisible. While the needs of others—nurses and non-medical staff—were considered, her needs were assumed not to exist, demonstrating the powerlessness of the junior doctors in this situation. If you cannot be seen, you cannot be helped.

The need for extra training and support for junior doctors during pandemics has been reported.[15] We echo this recommendation and would add that support could come from good leadership which recognises the challenges staff face, a feeling of being valued within a team and by addressing the practical and physical limitations junior doctors frequently experience, such as poor 'on

call' accommodation and access to regular meals. We suggest that employers often fail to recognise the limitations of individual coping strategies, both during crises such as the pandemic and in less unusual times.

Participants used various strategies to attempt to cope with working during COVID-19. Emotion-focused strategies such as crying were reported in our study although these strategies often appeared limited in usefulness. At times, the stoicism reported by participants in the current study verged on learnt helplessness, demonstrating that personal coping strategies alone are not enough and that coping is not guaranteed in a healthcare crisis when doctors are already stressed and distressed. Various individualised coping strategies have been suggested, including healthy eating, attending training, going to therapy, support networks[6] and making use of 'wobble rooms'.[15] However, Vera San Juan and colleagues[6] recognised that finding time for these activities might be difficult, particularly during a time of crisis.

Owens et al[30] state that if we are continually asking our HCPs to behave heroically in exceptional circumstances, we are inviting burnout. Indeed, it could be posited that encouraging such strategies places the responsibility for managing the unmanageable with individuals, rather than the system.[6] It is argued that, in our neo-liberal culture, responsibility for well-being is often placed on the individual, exonerating the state and systems for the well-being of workers.[35–37] This can be seen in the use of the term 'resilience', which places responsibility for managing the unmanageable on the shoulders of individuals, rather than organisations.[36 37] Therefore, in line with Vera San Juan et al,[6] we recommend a focus on organisational, rather than the personal, coping strategies. Those organisational strategies could, as seen in our findings, include flexibility and better organisational, managerial and peer support through teamwork and collaboration as well as addressing the practical workplace issues that could lead to HCPs feeling physically safe and cared for. Vulnerable junior doctors need organisational support especially, although not exclusively, during crises like COVID-19. However, the emphasis continues to be on the individual.[38 39]

Participants reported several potential positive impacts of working during the pandemic, which is a novel finding of this paper. These included working in more consistent teams. Vera San Juan and colleagues[6] have similarly reported that consistent teams are helpful for HCP, while inconsistent teams make it harder for junior doctors to seek support,[14] increasing stress and vulnerability to mental ill-health.[21] As such, we recommend that, where possible, policymakers consider the use of consistent teams for junior doctors going forward. The beneficial impact of a reduction in bureaucracy reported by one participant appears to be another novel finding. We would suggest any such reductions should be maintained after the pandemic ends, with a potential reduction in time pressures for junior—and senior—doctors as well as other healthcare workers.

Participants stated that some new supportive measures, such as rest hubs, had been put into place during COVID-19. Such spaces have been deemed helpful by other researchers,[7 10 15] although there are anecdotal reports that many of these spaces have now been closed as hospitalisations from COVID-19 reduce. In contrast, HCPs in other studies have reported that the extra strain on the system meant that there were fewer places than usual to shower, rest or relax with colleagues.[5 6]

In line with our findings, Vindrola-Padros et al[7] reported that there was extra signposting towards support during COVID-19; however, there was not often time to engage with this support. Additionally, it has been anecdotally reported that much of this support has been withdrawn now. This adds further weight to the notion that systemic, holistic changes are needed to support NHS staff, rather than focusing the responsibility for change on individuals.[6] We suggest that such limited responses from employers may have contributed to the feelings of resignation described by some of our participants.

### Limitations
This study has various strengths, including being the first qualitative paper (to our knowledge) to explore the experiences of junior doctors during COVID-19. Our data were collected during the pandemic, and we used in-depth, collaborative thematic analysis. However, despite these strengths, the paper has several limitations. We did not recruit these participants specifically to talk about the COVID-19 pandemic. Rather, the timing of the study meant that the topic arose naturally. As such, the interview guide could have been designed to ask participants more thoroughly about these experiences. Additionally, some of the junior doctors had more experience of working with patients with COVID-19 than others, meaning some participants are better represented in this paper than others. Furthermore, there is a notable gender disparity, with a higher proportion of female doctors taking part. More female (n=12) participants volunteered than males (n=3). The increased willingness of female participants to speak about their experiences may be associated with evidence indicating that female doctors are more likely to experience distress. Sadly, this group are also more likely to kill themselves.[1] The higher proportion of female participants may also reflect gendered help-seeking behaviour for mental ill-health, evidenced in the wider population,[39] as well as the fact that female doctors are more likely to take part in research than their male counterparts.[40] Finally, it should be reiterated that this was a purposive sample of particularly distressed junior doctors, although taken from a wider sample in which distress was widely reported, and so our findings are not intended to be generalised to all junior doctors.

## CONCLUSIONS AND RECOMMENDATIONS

We conclude that junior doctors working during the COVID-19 pandemic faced multiple stressors and used various coping mechanisms to deal with these, with greater or lesser degrees of success. Several unexpected benefits of this period arose, including new ways of working and additional support and camaraderie. We believe that the responsibility for alleviating the stress and distress of junior doctors working during times of stress lies with organisational employment issues and systemic workforce gaps, rather than with individuals. As such, we recommend system-wide changes, such as improved communication strategies, increased flexibility around home-based working, addressing the physical limitations of the working conditions many junior doctors experience and more supportive and compassionate leadership. Additionally, we suggest that, where possible, junior doctors are assigned to consistent teams, with the opportunity for appropriate psychological support where indicated.

### Author affiliations
[1]College of Medical and Dental Sciences, University of Birmingham, Birmingham, UK
[2]Research Department of Primary Care and Population Health, University College London, London, UK
[3]School of Medicine, University of Keele, Keele, UK
[4]Yorkshire Quality and Safety Research Group, Bradford Institute for Health Research, Bradford Teaching Hospitals NHS Foundation Trust, Bradford, UK
[5]Leeds Institute of Health Sciences, Faculty of Medicine and Health, University of Leeds, Leeds, UK
[6]Torbay Council, Torquay, UK
[7]University of Exeter, Exeter, UK
[8]Department of Organizational Psychology, Birkbeck, University of London, London, UK
[9]Department of Psychology and Mental Health, School of Medicine, University of Manchester, Manchester, UK
[10]Institute of Applied Health Research, University of Birmingham, Birmingham, UK

Acknowledgements The authors would like to thank all junior doctor participants who kindly gave their time to share their experiences of working in the National Health Service. We also wish to thank members of the Patient and Public Involvement and Engagement (PPIE) group who provided valuable input throughout the study.

Contributors All authors: substantial contributions to conception and design, and approval of final version to the published. RR, JS, CC-G, MB, AT, KT, AD and AT: acquisition, analysis or interpretation of data; and drafting the article or revising it critically for important intellectual content. JS acts as guarantor for this paper.

Competing interests None declared.

Patient and public involvement Patients and/or the public were involved in the design, or conduct, or reporting, or dissemination plans of this research. Refer to the Methods section for further details.

Patient consent for publication Not applicable.

Ethics approval Ethical approval was granted by the University of Birmingham and Health Research Authority (reference number: 19/HRA/6579).

Provenance and peer review Not commissioned; externally peer reviewed.

Data availability statement No data are available. This study has not received ethical approval to share confidential data with any third party other than the research team.

### ORCID iDs
Johanna Spiers http://orcid.org/0000-0002-3935-1997
Anna Kathryn Taylor http://orcid.org/0000-0002-8149-3841
Kevin Rui-Han Teoh http://orcid.org/0000-0002-6490-8208
Ruth Riley http://orcid.org/0000-0001-8774-5344

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
