## [Reviewer comments · BMJ Open]

ARTICLE DETAILS

TITLE (PROVISIONAL)	What challenges did junior doctors face whilst working during the COVID-19 pandemic? A qualitative study
AUTHORS	Spiers, Johanna; Buszewicz, Marta; Chew-Graham, Carolyn; Dunning, Alice; Taylor, Anna; Gopfert, Anya; Van Hove, Maria; Teoh, Kevin; Appleby, Louis; Martin, James; Riley, Ruth

VERSION 1 – REVIEW

REVIEWER	Lauren Alexander Trinity College, Psychiatry
REVIEW RETURNED	31-Aug-2021

GENERAL COMMENTS	This is an interesting study that lends insight into difficulties faced by those junior doctors suffering with mental health symptoms while working during the COVID-19 pandemic. It was an enjoyable and easy read! I have a few concerns and suggestions about the paper: 1. The authors state that the interview guide was informed by existing literature. This literature should be referenced in this part of the article.2. Authors make reference to a participants having access to "appropriate support" if suicidality is reported in the interview. What support is this?3. Reflexivity - perhaps I have misinterpreted this, but it sounds like some of the researchers are junior doctors themselves. Surely this is very relevant to how the results would be interpreted? How was the effect of pre-existing opinions mitigated?4. Table 1 needs to be populated with more quotes to support subthemes.5. The discussion is the weakest part of the article. While the results sets out the thematic analysis nicely, the discussion more or less repeats each subtheme and adds a reference from the literature to support each finding. The discussion should draw different results together, reflect on these in the context of existing literature, and reach new ideas that are not readily obvious to the reader from reading through the results section.6. I note that the study participants are junior doctors that reached a threshold in scales of mood and suicidality, and represents just 15 of a much larger sample. It is important that the findings do not generalise these results to all junior doctors. This is a study about the experience of working in COVID19 in junior doctors who reported certain symptoms. I think this should be made clearer, and perhaps developed a little more in the discussion.
---

REVIEWER	Christopher Wibberley Manchester Metropolitan Univ, Health, Psychology & Social Care
REVIEW RETURNED	06-Sep-2021

GENERAL COMMENTS

I looked forward to reading this paper, given the abstract provided. However, I was rather disappointed in the data presented and its analysis as part of this presentation. My interest peaked again, with the discussion though.

The data seems a little thin at times and I was often unconvinced about the match between the narrative of the presentation and the quotes used. This may be a result of the acknowledged limitation re

....

“We did not recruit these participants specifically to talk about the Covid-19 pandemic. Rather, the timing of the study meant that the topic arose naturally. As such, the interview guide could have been designed to ask participants more thoroughly about these experiences.”

Alternatively it may be an artefact of the collaborative approach to the analysis; whilst I can see that this may be driven by a desire to increase perceived validity, it may have weakened individual interpretive analysis. Additionally, given the critical realist approach espoused, I'm not sure about the fit between this (perception of mine) and the paradigm within which the authors report they are working.

This latter point also leads to another query I have - re the statement about reflexivity: a) should there be four positionality statements; and b) could the statement be less general or at least include some unpacking of preconceptions around individual versus corporate responsibility for managing stress etc. This would then strengthen links with discussion. However, this would suggest a deductive element to analysis - counter to the reflexivity statement; on the other hand it may lead to a more credible interpretive analysis.

Initially, given the phenomenological perspectives noted in the reflexivity statements I was wondering if more existential themes would be tackled, but I don't actually think this is a weakness within the analysis. There is a note made in discussion that “Participants also described existential fears about their safety and that of their loved ones”; however I'm not sure this statement is warranted by the data that is actually presented.

I'd like to stress that I really like the discussion - especially as it builds on the work of St Juan et al. I'm just not currently convinced about the link between the data presented (around COVID-19 and its link to the themes) and this discussion.

To some extent I don't 'need' the data per se to appreciate the arguments made in the discussion, I might prefer an opinion piece based on existing literature and the discussion. However, I accept that a data driven paper would be preferable, in which case I feel that the data side of the paper needs to be considered a little more.

I have provided further comments on individual aspects of the paper on the attached.

I am also aware that a paper has just been accepted by BMJopen around junior doctors and COVID-19 and the transition to clinical practice [the manuscript ID being bmjopen-2021-053423.R1} which may be available soon if revisions are to be made.

	The reviewer provided a marked copy with additional comments. Please contact the publisher for full details.
--	--

VERSION 1 – AUTHOR RESPONSE

Reviewer: 1 Dr Lauren Alexander, Trinity College

Comments to the Author:

This is an interesting study that lends insight into difficulties faced by those junior doctors suffering with mental health symptoms while working during the COVID-19 pandemic. It was an enjoyable and easy read!

Thank you for this lovely comment, we're glad you enjoyed the paper.

I have a few concerns and suggestions about the paper:

1. The authors state that the interview guide was informed by existing literature. This literature should be referenced in this part of the article.

We have now included this information – see page 11.

2. Authors make reference to participants' having access to "appropriate support" if suicidality is reported in the interview. What support is this?

We have now included this information – see page 11.

3. Reflexivity - perhaps I have misinterpreted this, but it sounds like some of the researchers are junior doctors themselves. Surely this is very relevant to how the results would be interpreted? How was the effect of pre-existing opinions mitigated?

While it is the case that some of our team are junior doctors, only one contributed to the analysis, and she commented on three transcripts rather than taking a lead in this role. We accept that this is a valid query, however, so have updated page 12 to make this clear.

4. Table 1 needs to be populated with more quotes to support subthemes.

Thanks for this comment. More quotes have now been added to the table on page 14.

5. The discussion is the weakest part of the article. While the results sets out the thematic analysis nicely, the discussion more or less repeats each subtheme and adds a reference from the literature to support each finding. The discussion should draw different results together, reflect on these in the context of existing literature, and reach new ideas that are not readily obvious to the reader from reading through the results section.

Thank you for this comment. We restructured our discussion section to look at our findings as a whole and have also added more references to further demonstrate the ways in which our findings build on existing work as well as agreeing with it. See pages 25-28.

6. I note that the study participants are junior doctors that reached a threshold in scales of mood and suicidality, and represents just 15 of a much larger sample. It is important that the findings do not generalise these results to all junior doctors. This is a study about the experience of working in COVID-19 in junior doctors who reported certain symptoms. I think this should be made clearer, and perhaps developed a little more in the discussion.

This is a good point, thank you. Please see amended text on pages 10, 11, 25-28.

Reviewer: 2 Dr Christopher Wibberley, Manchester Metropolitan Univ

Comments to the Author:

*** Please find additional comments from this reviewer in the attached file ***

I looked forward to reading this paper, given the abstract provided. However, I was rather disappointed in the data presented and its analysis as part of this presentation. My interest peaked again, with the discussion though.

The data seems a little thin at times and I was often unconvinced about the match between the narrative of the presentation and the quotes used. This may be a result of the acknowledged limitation re “We did not recruit these participants specifically to talk about the Covid-19 pandemic. Rather, the timing of the study meant that the topic arose naturally. As such, the interview guide could have been designed to ask participants more thoroughly about these experiences.”

Alternatively it may be an artefact of the collaborative approach to the analysis; whilst I can see that this may be driven by a desire to increase perceived validity, it may have weakened individual interpretive analysis. Additionally, given the critical realist approach espoused, I’m not sure about the fit between this (perception of mine) and the paradigm within which the authors report they are working.

Thank you for this thoughtful comment. We have added more detail about our analytic approach to page 12, deepened the analysis on pages 15-25, adjusted some theme titles and added more data on page 19. We hope that this addresses your concerns.

This latter point also leads to another query I have - re the statement about reflexivity: a) should there be four positionality statements; and b) could the statement be less general or at least include some unpacking of preconceptions around individual versus corporate responsibility for managing stress etc. This would then strengthen links with discussion. However, this would suggest a deductive element to analysis - counter to the reflexivity statement; on the other hand it may lead to a more credible interpretive analysis.

Thank you for this comment. Please see amended text on pages 12 and 13.

Initially, given the phenomenological perspectives noted in the reflexivity statements I was wondering if more existential themes would be tackled, but I don’t actually think this is a weakness within the analysis. There is a note made in discussion that “Participants also described existential

fears about their safety and that of their loved ones”; however I’m not sure this statement is warranted by the data that is actually presented.

Following comments from reviewer 1, this paragraph has now been removed from the discussion section.

I’d like to stress that I really like the discussion - especially as it builds on the work of St Juan et al. I’m just not currently convinced about the link between the data presented (around COVID-19 and its link to the themes) and this discussion.

Thank you for this kind comment. We hope that the amendments we have made to the discussion based on reviewer 1’s comments as well as our newly deepened analysis strengthens that link.

To some extent I don’t ‘need’ the data per se to appreciate the arguments made in the discussion, I might prefer an opinion piece based on existing literature and the discussion. However, I accept that a data driven paper would be preferable, in which case I feel that the data side of the paper needs to be considered a little more.

We thank the reviewer for this comment. Our study, involving in-depth interviews with junior doctors, was always designed to provide qualitative data to inform our position. We hope that we have now covered the concerns of both reviewers.

I have provided further comments on individual aspects of the paper on the attached.

I am also aware that a paper has just been accepted by BMJopen around junior doctors and COVID-19 and the transition to clinical practice [the manuscript ID being bmjopen-2021-053423.R1] which may be available soon if revisions are to be made.

Thank you; I emailed the BMJ Open about this paper, but it is still in production and so we could not access it.

What was the gender split for the target group, as identified through the DASS-21 tool?

Thanks for this valid question. We have included this info on page 11.

Newman et al regard moral injury as: ‘making difficult decisions against moral judgement’ which seems to be a slightly different interpretation.

We have removed the reference to Newman on page 26.

This is I think interesting, and probably little considered in the literature Could more be made of this.

Thanks for this comment – see additional lines on pages 27 and 28.

The participants in this study didn't use the term 'wobble rooms' as reported in the 'findings' so??

See amended text on page 29.

VERSION 2 – REVIEW

REVIEWER	Christopher Wibberley Manchester Metropolitan Univ, Health, Psychology & Social Care
REVIEW RETURNED	28-Oct-2021

GENERAL COMMENTS	I made the following comments about the original submission The data seems a little thin at times and I was often unconvinced about the match between the narrative of the presentation and the quotes used. THIS HAS NOW BEEN ADDRESSED re the statement about reflexivity: a) should there be four positionality statements; and b) could the statement be less general or at least include some unpacking of preconceptions around individual versus corporate responsibility for managing stress etc. This would then strengthen links with discussion. THIS HAS NOW BEEN ADDRESSED I'm just not currently convinced about the link between the data presented (around COVID-19 and its link to the themes) and this discussion. THE DATA PRESENTED HAS NOW BEEN ADAPTED TO ADDRESS THIS COMMENT, WITH ORGANISATIONAL SUPPORT / THE LACK OF IT ... BEING CLEARER WITHIN DATA PRESENTATION. Additionally.... Re this submission I have highlighted the odd phrase / word (see attached) that should be considered in a final proof read. The reviewer provided a marked copy with additional comments. Please contact the publisher for full details.
--